# Learning to Prune Deep Neural Networks via Layer-wise Optimal Brain Surgeon

**Xin Dong**
Nanyang Technological University, Singapore
n1503521a@e.ntu.edu.sg

**Shangyu Chen**
Nanyang Technological University, Singapore
schen025@e.ntu.edu.sg

**Sinno Jialin Pan**
Nanyang Technological University, Singapore
sinnopan@ntu.edu.sg

## Abstract

How to develop slim and accurate deep neural networks has become crucial for real-world applications, especially for those employed in embedded systems. Though previous work along this research line has shown some promising results, most existing methods either fail to significantly compress a well-trained deep network or require a heavy retraining process for the pruned deep network to re-boost its prediction performance. In this paper, we propose a new layer-wise pruning method for deep neural networks. In our proposed method, parameters of each individual layer are pruned independently based on second order derivatives of a layer-wise error function with respect to the corresponding parameters. We prove that the final prediction performance drop after pruning is bounded by a linear combination of the reconstructed errors caused at each layer. By controlling layer-wise errors properly, one only needs to perform a light retraining process on the pruned network to resume its original prediction performance. We conduct extensive experiments on benchmark datasets to demonstrate the effectiveness of our pruning method compared with several state-of-the-art baseline methods. Codes of our work are released at: https://github.com/csyhhu/L-OBS.

## 1 Introduction

Intuitively, deep neural networks [1] can approximate predictive functions of arbitrary complexity well when they are of a huge amount of parameters, i.e., a lot of layers and neurons. In practice, the size of deep neural networks has been being tremendously increased, from LeNet-5 with less than 1M parameters [2] to VGG-16 with 133M parameters [3]. Such a large number of parameters not only make deep models memory intensive and computationally expensive, but also urge researchers to dig into redundancy of deep neural networks. On one hand, in neuroscience, recent studies point out that there are significant redundant neurons in human brain, and memory may have relation with vanishment of specific synapses [4]. On the other hand, in machine learning, both theoretical analysis and empirical experiments have shown the evidence of redundancy in several deep models [5, 6]. Therefore, it is possible to compress deep neural networks without or with little loss in prediction by pruning parameters with carefully designed criteria.

However, finding an optimal pruning solution is NP-hard because the search space for pruning is exponential in terms of parameter size. Recent work mainly focuses on developing efficient algorithms to obtain a near-optimal pruning solution [7, 8, 9, 10, 11]. A common idea behind most exiting approaches is to select parameters for pruning based on certain criteria, such as increase in training error, magnitude of the parameter values, etc. As most of the existing pruning criteria are

designed heuristically, there is no guarantee that prediction performance of a deep neural network can be preserved after pruning. Therefore, a time-consuming retraining process is usually needed to boost the performance of the trimmed neural network.

Instead of consuming efforts on a whole deep network, a layer-wise pruning method, Net-Trim, was proposed to learn sparse parameters by minimizing reconstructed error for each individual layer [6]. A theoretical analysis is provided that the overall performance drop of the deep network is bounded by the sum of reconstructed errors for each layer. In this way, the pruned deep network has a theoretical guarantee on its error. However, as Net-Trim adopts $\ell_1$-norm to induce sparsity for pruning, it fails to obtain high compression ratio compared with other methods [9, 11].

In this paper, we propose a new layer-wise pruning method for deep neural networks, aiming to achieve the following three goals: 1) For each layer, parameters can be highly compressed after pruning, while the reconstructed error is small. 2) There is a theoretical guarantee on the overall prediction performance of the pruned deep neural network in terms of reconstructed errors for each layer. 3) After the deep network is pruned, only a light retraining process is required to resume its original prediction performance.

To achieve our first goal, we borrow an idea from some classic pruning approaches for shallow neural networks, such as optimal brain damage (OBD) [12] and optimal brain surgeon (OBS) [13]. These classic methods approximate a change in the error function via functional Taylor Series, and identify unimportant weights based on second order derivatives. Though these approaches have proven to be effective for shallow neural networks, it remains challenging to extend them for deep neural networks because of the high computational cost on computing second order derivatives, i.e., the inverse of the Hessian matrix over all the parameters. In this work, as we restrict the computation on second order derivatives w.r.t. the parameters of each individual layer only, i.e., the Hessian matrix is only over parameters for a specific layer, the computation becomes tractable. Moreover, we utilize characteristics of back-propagation for fully-connected layers in well-trained deep networks to further reduce computational complexity of the inverse operation of the Hessian matrix.

To achieve our second goal, based on the theoretical results in [6], we provide a proof on the bound of performance drop before and after pruning in terms of the reconstructed errors for each layer. With such a layer-wise pruning framework using second-order derivatives for trimming parameters for each layer, we empirically show that after significantly pruning parameters, there is only a little drop of prediction performance compared with that before pruning. Therefore, only a light retraining process is needed to resume the performance, which achieves our third goal.

The contributions of this paper are summarized as follows. 1) We propose a new layer-wise pruning method for deep neural networks, which is able to significantly trim networks and preserve the prediction performance of networks after pruning with a theoretical guarantee. In addition, with the proposed method, a time-consuming retraining process for re-boosting the performance of the pruned network is waived. 2) We conduct extensive experiments to verify the effectiveness of our proposed method compared with several state-of-the-art approaches.

## 2 Related Works and Preliminary

Pruning methods have been widely used for model compression in early neural networks [7] and modern deep neural networks [6, 8, 9, 10, 11]. In the past, with relatively small size of training data, pruning is crucial to avoid overfitting. Classical methods include OBD and OBS. These methods aim to prune parameters with the least increase of error approximated by second order derivatives. However, computation of the Hessian inverse over all the parameters is expensive. In OBD, the Hessian matrix is restricted to be a diagonal matrix to make it computationally tractable. However, this approach implicitly assumes parameters have no interactions, which may hurt the pruning performance. Different from OBD, OBS makes use of the full Hessian matrix for pruning. It obtains better performance while is much more computationally expensive even using Woodbury matrix identity [14], which is an iterative method to compute the Hessian inverse. For example, using OBS on VGG-16 naturally requires to compute inverse of the Hessian matrix with a size of $133\text{M} \times 133\text{M}$.

Regarding pruning for modern deep models, Han et al. [9] proposed to delete unimportant parameters based on magnitude of their absolute values, and retrain the remaining ones to recover the original prediction performance. This method achieves considerable compression ratio in practice. However,

as pointed out by pioneer research work [12, 13], parameters with low magnitude of their absolute values can be necessary for low error. Therefore, magnitude-based approaches may eliminate wrong parameters, resulting in a big prediction performance drop right after pruning, and poor robustness before retraining [15]. Though some variants have tried to find better magnitude-based criteria [16, 17], the significant drop of prediction performance after pruning still remains. To avoid pruning wrong parameters, Guo et al. [11] introduced a mask matrix to indicate the state of network connection for dynamically pruning after each gradient decent step. Jin et al. [18] proposed an iterative hard thresholding approach to re-activate the pruned parameters after each pruning phase.

Besides Net-trim, which is a layer-wise pruning method discussed in the previous section, there is some other work proposed to induce sparsity or low-rank approximation on certain layers for pruning [19, 20]. However, as the $\ell_0$-norm or the $\ell_1$-norm sparsity-induced regularization term increases difficulty in optimization, the pruned deep neural networks using these methods either obtain much smaller compression ratio [6] compared with direct pruning methods or require retraining of the whole network to prevent accumulation of errors [10].

**Optimal Brain Surgeon** As our proposed layer-wise pruning method is an extension of OBS on deep neural networks, we briefly review the basic of OBS here. Consider a network in terms of parameters $\mathbf{w}$ trained to a local minimum in error. The functional Taylor series of the error w.r.t. $\mathbf{w}$ is: $\delta E = \left(\frac{\partial E}{\partial \mathbf{w}}\right)^\top \delta \mathbf{w} + \frac{1}{2}\delta \mathbf{w}^\top \mathbf{H}\delta \mathbf{w} + O\left(\|\delta \mathbf{w}\|^3\right)$, where $\delta$ denotes a perturbation of a corresponding variable, $\mathbf{H} \equiv \partial^2 E / \partial \mathbf{w}^2 \in \mathbb{R}^{m \times m}$ is the Hessian matrix, where $m$ is the number of parameters, and $O(\|\delta \mathbf{\Theta}_l\|^3)$ is the third and all higher order terms. For a network trained to a local minimum in error, the first term vanishes, and the term $O(\|\delta \mathbf{\Theta}_l\|^3)$ can be ignored. In OBS, the goal is to set one of the parameters to zero, denoted by $w_q$ (scalar), to minimize $\delta E$ in each pruning iteration. The resultant optimization problem is written as follows,

$$\min_q \frac{1}{2}\delta \mathbf{w}^\top \mathbf{H}\delta \mathbf{w}, \quad \text{s.t.} \quad \mathbf{e}_q^\top \delta \mathbf{w} + \mathbf{w}_q = 0, \tag{1}$$

where $\mathbf{e}_q$ is the unit selecting vector whose $q$-th element is 1 and otherwise 0. As shown in [21], the optimization problem (1) can be solved by the Lagrange multipliers method. Note that a computation bottleneck of OBS is to calculate and store the non-diagonal Hesssian matrix and its inverse, which makes it impractical on pruning deep models which are usually of a huge number of parameters.

# 3 Layer-wise Optimal Brain Surgeon

## 3.1 Problem Statement

Given a training set of $n$ instances, $\{(\mathbf{x}_j, y_j)\}_{j=1}^n$, and a well-trained deep neural network of $L$ layers (excluding the input layer)[1]. Denote the input and the output of the whole deep neural network by $\mathbf{X} = [\mathbf{x}_1, ..., \mathbf{x}_n] \in \mathbb{R}^{d \times n}$ and $\mathbf{Y} \in \mathbb{R}^{n \times 1}$, respectively. For a layer $l$, we denote the input and output of the layer by $\mathbf{Y}^{l-1} = [\mathbf{y}_1^{l-1}, ..., \mathbf{y}_n^{l-1}] \in \mathbb{R}^{m_{l-1} \times n}$ and $\mathbf{Y}^l = [\mathbf{y}_1^l, ..., \mathbf{y}_n^l] \in \mathbb{R}^{m_l \times n}$, respectively, where $\mathbf{y}_i^l$ can be considered as a representation of $\mathbf{x}_i$ in layer $l$, and $\mathbf{Y}^0 = \mathbf{X}$, $\mathbf{Y}^L = \mathbf{Y}$, and $m_0 = d$. Using one forward-pass step, we have $\mathbf{Y}^l = \sigma(\mathbf{Z}^l)$, where $\mathbf{Z}^l = \mathbf{W}_l^\top \mathbf{Y}^{l-1}$ with $\mathbf{W}_l \in \mathbb{R}^{m_{l-1} \times m_l}$ being the matrix of parameters for layer $l$, and $\sigma(\cdot)$ is the activation function. For convenience in presentation and proof, we define the activation function $\sigma(\cdot)$ as the rectified linear unit (ReLU) [22]. We further denote by $\mathbf{\Theta}_l \in \mathbb{R}^{m_{l-1} m_l \times 1}$ the vectorization of $\mathbf{W}_l$. For a well-trained neural network, $\mathbf{Y}^l$, $\mathbf{Z}^l$ and $\mathbf{\Theta}_l^*$ are all fixed matrixes and contain most information of the neural network. The goal of pruning is to set the values of some elements in $\mathbf{\Theta}_l$ to be zero.

## 3.2 Layer-Wise Error

During layer-wise pruning in layer $l$, the input $\mathbf{Y}^{l-1}$ is fixed as the same as the well-trained network. Suppose we set the $q$-th element of $\mathbf{\Theta}_l$, denoted by $\mathbf{\Theta}_{l_{[q]}}$, to be zero, and get a new parameter vector, denoted by $\hat{\mathbf{\Theta}}_l$. With $\mathbf{Y}^{l-1}$, we obtain a new output for layer $l$, denoted by $\hat{\mathbf{Y}}^l$. Consider the root of

mean square error between $\hat{\mathbf{Y}}^l$ and $\mathbf{Y}^l$ over the whole training data as the layer-wise error:

$$\varepsilon^l = \sqrt{\frac{1}{n} \sum_{j=1}^n \left( (\hat{\mathbf{y}}_j^l - \mathbf{y}_j^l)^\top (\hat{\mathbf{y}}_j^l - \mathbf{y}_j^l) \right)} = \frac{1}{\sqrt{n}} \| \hat{\mathbf{Y}}^l - \mathbf{Y}^l \|_F, \tag{2}$$

where $\| \cdot \|_F$ is the Frobenius Norm. Note that for any single parameter pruning, one can compute its error $\varepsilon_q^l$, where $1 \le q \le m_{l-1} m_l$, and use it as a pruning criterion. This idea has been adopted by some existing methods [15]. However, in this way, for each parameter at each layer, one has to pass the whole training data once to compute its error measure, which is very computationally expensive. A more efficient approach is to make use of the second order derivatives of the error function to help identify importance of each parameter.

We first define an error function $E(\cdot)$ as

$$E^l = E(\hat{\mathbf{Z}}^l) = \frac{1}{n} \left\| \hat{\mathbf{Z}}^l - \mathbf{Z}^l \right\|_F^2, \tag{3}$$

where $\mathbf{Z}^l$ is outcome of the weighted sum operation right before performing the activation function $\sigma(\cdot)$ at layer $l$ of the well-trained neural network, and $\hat{\mathbf{Z}}^l$ is outcome of the weighted sum operation after pruning at layer $l$ . Note that $\mathbf{Z}^l$ is considered as the desired output of layer $l$ before activation. The following lemma shows that the layer-wise error is bounded by the error defined in (3).

**Lemma 3.1.** *With the error function* (3) *and* $\mathbf{Y}^l = \sigma(\mathbf{Z}^l)$, *the following holds:* $\varepsilon^l \le \sqrt{E(\hat{\mathbf{Z}}^l)}$.

Therefore, to find parameters whose deletion (set to be zero) minimizes (2) can be translated to find parameters those deletion minimizes the error function (3). Following [12, 13], the error function can be approximated by functional Taylor series as follows,

$$E(\hat{\mathbf{Z}}^l) - E(\mathbf{Z}^l) = \delta E^l = \left( \frac{\partial E^l}{\partial \boldsymbol{\Theta}_l} \right)^\top \delta \boldsymbol{\Theta}_l + \frac{1}{2} \delta \boldsymbol{\Theta}_l^\top \mathbf{H}_l \delta \boldsymbol{\Theta}_l + O\left( \|\delta \boldsymbol{\Theta}_l\|^3 \right), \tag{4}$$

where $\delta$ denotes a perturbation of a corresponding variable, $\mathbf{H}_l \equiv \partial^2 E^l / \partial \boldsymbol{\Theta}_l^2$ is the Hessian matrix w.r.t. $\boldsymbol{\Theta}_l$, and $O(\|\delta \boldsymbol{\Theta}_l\|^3)$ is the third and all higher order terms. It can be proven that with the error function defined in (3), the first (linear) term $\frac{\partial E^l}{\partial \boldsymbol{\Theta}_l}\big|_{\boldsymbol{\Theta}_l = \boldsymbol{\Theta}_l^*}$ and $O(\|\delta \boldsymbol{\Theta}_l\|^3)$ are equal to 0.

Suppose every time one aims to find a parameter $\boldsymbol{\Theta}_{l_{[q]}}$ to set to be zero such that the change $\delta E^l$ is minimal. Similar to OBS, we can formulate it as the following optimization problem:

$$\min_q \frac{1}{2} \delta \boldsymbol{\Theta}_l^\top \mathbf{H}_l \delta \boldsymbol{\Theta}_l, \quad \text{s.t.} \quad \mathbf{e}_q^\top \delta \boldsymbol{\Theta}_l + \boldsymbol{\Theta}_{l_{[q]}} = 0, \tag{5}$$

where $\mathbf{e}_q$ is the unit selecting vector whose $q$-th element is 1 and otherwise 0. By using the Lagrange multipliers method as suggested in [21], we obtain the closed-form solutions of the optimal parameter pruning and the resultant minimal change in the error function as follows,

$$\delta \boldsymbol{\Theta}_l = -\frac{\boldsymbol{\Theta}_{l_{[q]}}}{[\mathbf{H}_l^{-1}]_{qq}} \mathbf{H}_l^{-1} \mathbf{e}_q, \quad \text{and} \quad L_q = \delta E^l = \frac{1}{2} \frac{(\boldsymbol{\Theta}_{l_{[q]}})^2}{[\mathbf{H}_l^{-1}]_{qq}}. \tag{6}$$

Here $L_q$ is referred to as the sensitivity of parameter $\boldsymbol{\Theta}_{l_{[q]}}$. Then we select parameters to prune based on their sensitivity scores instead of their magnitudes. As mentioned in section 2, magnitude-based criteria which merely consider the numerator in (6) is a poor estimation of sensitivity of parameters. Moreover, in (6), as the inverse Hessian matrix over the training data is involved, it is able to capture data distribution when measuring sensitivities of parameters.

After pruning the parameter, $\boldsymbol{\Theta}_{l_{[q]}}$, with the smallest sensitivity, the parameter vector is updated via $\hat{\boldsymbol{\Theta}}_l = \boldsymbol{\Theta}_l + \delta \boldsymbol{\Theta}_l$. With Lemma 3.1 and (6), we have that the layer-wise error for layer $l$ is bounded by

$$\varepsilon_q^l \le \sqrt{E(\hat{\mathbf{Z}}^l)} = \sqrt{E(\hat{\mathbf{Z}}^l) - E(\mathbf{Z}^l)} = \sqrt{\delta E^l} = \frac{|\boldsymbol{\Theta}_{l_{[q]}}|}{\sqrt{2[\mathbf{H}_l^{-1}]_{qq}}}. \tag{7}$$

Note that first equality is obtained because of the fact that $E(\mathbf{Z}^l) = 0$. It is worth to mention that though we merely focus on layer $l$, the Hessian matrix is still a square matrix with size of $m_{l-1} m_l \times m_{l-1} m_l$. However, we will show how to significantly reduce the computation of $\mathbf{H}_l^{-1}$ for each layer in Section 3.4.

### 3.3 Layer-Wise Error Propagation and Accumulation

So far, we have shown how to prune parameters for each layer, and estimate their introduced errors independently. However, our aim is to control the consistence of the network's final output $\mathbf{Y}^L$ before and after pruning. To do this, in the following, we show how the layer-wise errors propagate to final output layer, and the accumulated error over multiple layers will **not** explode.

**Theorem 3.2.** *Given a pruned deep network via layer-wise pruning introduced in Section 3.2, each layer has its own layer-wise error $\varepsilon^l$ for $1 \leq l \leq L$, then the accumulated error of ultimate network output $\tilde{\varepsilon}^L = \frac{1}{\sqrt{n}} \|\tilde{\mathbf{Y}}^L - \mathbf{Y}^L\|_F$ obeys:*

$$\tilde{\varepsilon}^L \leq \sum_{k=1}^{L-1} \left( \prod_{l=k+1}^{L} \|\hat{\mathbf{\Theta}}_l\|_F \sqrt{\delta E^k} \right) + \sqrt{\delta E^L}, \tag{8}$$

*where $\tilde{\mathbf{Y}}^l = \sigma(\hat{\mathbf{W}}_l^\top \tilde{\mathbf{Y}}^{l-1})$, for $2 \leq l \leq L$ denotes 'accumulated pruned output' of layer $l$, and $\tilde{\mathbf{Y}}^1 = \sigma(\hat{\mathbf{W}}_1^\top \mathbf{X})$.*

Theorem 3.2 shows that: 1) Layer-wise error for a layer $l$ will be scaled by continued multiplication of parameters' Frobenius Norm over the following layers when it propagates to final output, i.e., the $L-l$ layers after the $l$-th layer; 2) The final error of ultimate network output is bounded by the weighted sum of layer-wise errors. The proof of Theorem 3.2 can be found in Appendix.

Consider a general case with (6) and (8): parameter $\mathbf{\Theta}_{l_{[q]}}$ who has the smallest sensitivity in layer $l$ is pruned by the $i$-th pruning operation, and this finally adds $\prod_{k=l+1}^{L} \|\hat{\Theta}_k\|_F \sqrt{\delta E^l}$ to the ultimate network output error. It is worth to mention that although it seems that the layer-wise error is scaled by a quite large product factor, $S_l = \prod_{k=l+1}^{L} \|\hat{\Theta}_k\|_F$ when it propagates to the final layer, this scaling is still tractable in practice because ultimate network output is also scaled by the same product factor compared with the output of layer $l$. For example, we can easily estimate the norm of ultimate network output via, $\|\mathbf{Y}^L\|_F \approx S_1 \|\mathbf{Y}^1\|_F$. If one pruning operation in the 1st layer causes the layer-wise error $\sqrt{\delta E^1}$, then the **relative** ultimate output error is

$$\xi_r^L = \frac{\|\tilde{\mathbf{Y}}^L - \mathbf{Y}^L\|_F}{\|\mathbf{Y}^L\|_F} \approx \frac{\sqrt{\delta E^1}}{\|\frac{1}{n}\mathbf{Y}^1\|_F}.$$

Thus, we can see that even $S_1$ may be quite large, the relative ultimate output error would still be about $\sqrt{\delta E^1}/\|\frac{1}{n}\mathbf{Y}^1\|_F$ which is controllable in practice especially when most of modern deep networks adopt maxout layer [23] as ultimate output. Actually, $S_0$ is called as network gain representing the ratio of the magnitude of the network output to the magnitude of the network input.

### 3.4 The Proposed Algorithm

#### 3.4.1 Pruning on Fully-Connected Layers

To selectively prune parameters, our approach needs to compute the inverse Hessian matrix at each layer to measure the sensitivities of each parameter of the layer, which is still computationally expensive though tractable. In this section, we present an efficient algorithm that can reduce the size of the Hessian matrix and thus speed up computation on its inverse.

For each layer $l$, according to the definition of the error function used in Lemma 3.1, the first derivative of the error function with respect to $\hat{\mathbf{\Theta}}_l$ is $\frac{\partial E^l}{\partial \mathbf{\Theta}_l} = -\frac{1}{n} \sum_{j=1}^{n} \frac{\partial z_j^l}{\partial \mathbf{\Theta}_l} (\hat{\mathbf{z}}_j^l - \mathbf{z}_j^l)$, where $\hat{\mathbf{z}}_j^l$ and $\mathbf{z}_j^l$ are the $j$-th columns of the matrices $\hat{\mathbf{Z}}^l$ and $\mathbf{Z}^l$, respectively, and the Hessian matrix is defined as:
$\mathbf{H}_l \equiv \frac{\partial^2 E^l}{\partial(\mathbf{\Theta}_l)^2} = \frac{1}{n} \sum_{j=1}^{n} \left( \frac{\partial z_j^l}{\partial \mathbf{\Theta}_l} \left( \frac{\partial z_j^l}{\partial \mathbf{\Theta}_l} \right)^\top - \frac{\partial^2 z_j^l}{\partial(\mathbf{\Theta}_l)^2} (\hat{\mathbf{z}}_j^l - \mathbf{z}_j^l)^\top \right)$. Note that for most cases $\hat{\mathbf{z}}_j^l$ is quite close to $\mathbf{z}_j^l$, we simply ignore the term containing $\hat{\mathbf{z}}_j^l - \mathbf{z}_j^l$. Even in the late-stage of pruning when this difference is not small, we can still ignore the corresponding term [13]. For layer $l$ that has $m_l$ output units, $\mathbf{z}_j^l = [z_{1j}^l, \ldots, z_{m_l j}^l]$, the Hessian matrix can be calculated via

$$\mathbf{H}_l = \frac{1}{n} \sum_{j=1}^{n} \mathbf{H}_l^j = \frac{1}{n} \sum_{j=1}^{n} \sum_{i=1}^{m_l} \frac{\partial z_{ij}^l}{\partial \mathbf{\Theta}_l} \left( \frac{\partial z_{ij}^l}{\partial \mathbf{\Theta}_l} \right)^\top, \tag{9}$$

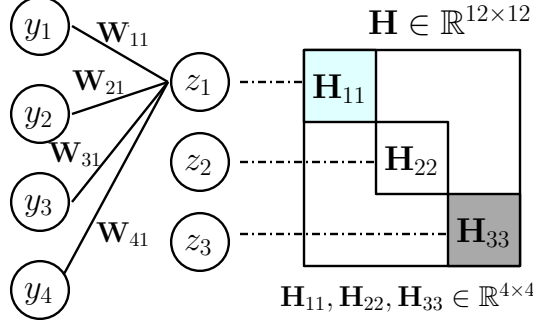

Figure 1: Illustration of shape of Hessian. For feed-forward neural networks, unit $z_1$ gets its activation via forward propagation: $\mathbf{z} = \mathbf{W}^\top \mathbf{y}$, where $\mathbf{W} \in \mathbb{R}^{4 \times 3}$, $\mathbf{y} = [y_1, y_2, y_3, y_4]^\top \in \mathbb{R}^{4 \times 1}$, and $\mathbf{z} = [z_1, z_2, z_3]^\top \in \mathbb{R}^{3 \times 1}$. Then the Hessian matrix of $z_1$ w.r.t. all parameters is denoted by $\mathbf{H}^{[z_1]}$. As illustrated in the figure, $\mathbf{H}^{[z_1]}$'s elements are zero except for those corresponding to $\mathbf{W}_{*1}$ (the 1st column of $\mathbf{W}$), which is denoted by $\mathbf{H}_{11}$. $\mathbf{H}^{[z_2]}$ and $\mathbf{H}^{[z_3]}$ are similar. More importantly, $\mathbf{H}^{-1} = \operatorname{diag}(\mathbf{H}_{11}^{-1}, \mathbf{H}_{22}^{-1}, \mathbf{H}_{33}^{-1})$, and $\mathbf{H}_{11} = \mathbf{H}_{22} = \mathbf{H}_{33}$. As a result, one only needs to compute $\mathbf{H}_{11}^{-1}$ to obtain $\mathbf{H}^{-1}$ which significantly reduces computational complexity.

where the Hessian matrix for a single instance $j$ at layer $l$, $\mathbf{H}_l^j$, is a block diagonal square matrix of the size $m_{l-1} \times m_l$. Specifically, the gradient of the first output unit $z_{1j}^l$ w.r.t. $\boldsymbol{\Theta}_l$ is $\frac{\partial z_{1j}^l}{\partial \boldsymbol{\Theta}_l} = \left[ \frac{\partial z_{1j}^l}{\partial \mathbf{w}_1}, \ldots, \frac{\partial z_{1j}^l}{\partial \mathbf{w}_{m_l}} \right]$, where $\mathbf{w}_i$ is the $i$-th column of $\mathbf{W}_l$. As $z_{1j}^l$ is the layer output before activation function, its gradient is simply to calculate, and more importantly all output units's gradients are equal to the layer input: $\frac{\partial z_{ij}^l}{\partial \mathbf{w}_k} = \mathbf{y}_j^{l-1}$ if $k = i$, otherwise $\frac{\partial z_{ij}^l}{\partial \mathbf{w}_k} = 0$. An illustrated example is shown in Figure 1, where we ignore the scripts $j$ and $l$ for simplicity in presentation.

It can be shown that the block diagonal square matrix $\mathbf{H}_l^j$'s diagonal blocks $\mathbf{H}_{l_{ii}}^j \in \mathbb{R}^{m_{l-1} \times m_{l-1}}$, where $1 \le i \le m_l$, are all equal to $\boldsymbol{\psi}_l^j = \mathbf{y}_j^{l-1}(\mathbf{y}_j^{l-1})^\top$, and the inverse Hessian matrix $\mathbf{H}_l^{-1}$ is also a block diagonal square matrix with its diagonal blocks being $(\frac{1}{n} \sum_{j=1}^n \boldsymbol{\psi}_l^j)^{-1}$. In addition, normally $\boldsymbol{\Psi}^l = \frac{1}{n} \sum_{j=1}^n \boldsymbol{\psi}_l^j$ is degenerate and its pseudo-inverse can be calculated recursively via Woodbury matrix identity [13]:

$$\left(\boldsymbol{\Psi}_{j+1}^l\right)^{-1} = \left(\boldsymbol{\Psi}_j^l\right)^{-1} - \frac{\left(\boldsymbol{\Psi}_j^l\right)^{-1} \mathbf{y}_j^{l-1} \left(\mathbf{y}_j^{l-1}\right)^\top \left(\boldsymbol{\Psi}_j^l\right)^{-1}}{n + \left(\mathbf{y}_{j+1}^{l-1}\right)^\top \left(\boldsymbol{\Psi}_j^l\right)^{-1} \mathbf{y}_{j+1}^{l-1}},$$

where $\boldsymbol{\Psi}_t^l = \frac{1}{t} \sum_{j=1}^t \boldsymbol{\psi}_l^j$ with $\left(\boldsymbol{\Psi}_0^l\right)^{-1} = \alpha \mathbf{I}$, $\alpha \in [10^4, 10^8]$, and $\left(\boldsymbol{\Psi}^l\right)^{-1} = \left(\boldsymbol{\Psi}_n^l\right)^{-1}$. The size of $\boldsymbol{\Psi}^l$ is then reduced to $m_{l-1}$, and the computational complexity of calculating $\mathbf{H}_l^{-1}$ is $O\left(nm_{l-1}^2\right)$.

To make the estimated minimal change of the error function optimal in (6), the layer-wise Hessian matrices need to be exact. Since the layer-wise Hessian matrices only depend on the corresponding layer inputs, they are always able to be exact even after several pruning operations. The only parameter we need to control is the layer-wise error $\varepsilon^l$. Note that there may be a "pruning inflection point" after which layer-wise error would drop dramatically. In practice, user can incrementally increase the size of pruned parameters based on the sensitivity $L_q$, and make a trade-off between the pruning ratio and the performance drop to set a proper tolerable error threshold or pruning ratio.

The procedure of our pruning algorithm for a fully-connected layer $l$ is summarized as follows.

Step 1: Get layer input $\mathbf{y}^{l-1}$ from a well-trained deep network.

Step 2: Calculate the Hessian matrix $\mathbf{H}_{l_{ii}}$, for $i = 1, \ldots, m_l$, and its pseudo-inverse over the dataset, and get the whole pseudo-inverse of the Hessian matrix.

Step 3: Compute optimal parameter change $\delta\boldsymbol{\Theta}_l$ and the sensitivity $L_q$ for each parameter at layer $l$. Set tolerable error threshold $\epsilon$.

Step 4: Pick up parameters $\Theta_{l_{[q]}}$'s with the smallest sensitivity scores.

Step 5: If $\sqrt{L_q} \le \epsilon$, prune the parameter $\Theta_{l_{[q]}}$'s and get new parameter values via $\hat{\Theta}_l = \Theta_l + \delta\Theta_l$, then repeat Step 4; otherwise stop pruning.

### 3.4.2 Pruning on Convolutional Layers

It is straightforward to generalize our method to a convolutional layer and its variants if we vectorize filters of each channel and consider them as special fully-connected layers that have multiple inputs (patches) from a single instance. Consider a vectorized filter $\mathbf{w}_i$ of channel $i$, $1 \le i \le m_l$, it acts similarly to parameters which are connected to the same output unit in a fully-connected layer. However, the difference is that for a single input instance $j$, every filter step of a sliding window across of it will extract a patch $C_{j_n}$ from the input volume. Similarly, each pixel $z_{ij_n}^l$ in the 2-dimensional activation map that gives the response to each patch corresponds to one output unit in a fully-connected layer. Hence, for convolutional layers, (9) is generalized as $\mathbf{H}_l = \frac{1}{n} \sum_{j=1}^{n} \sum_{i=1}^{m_l} \sum_{j_n} \frac{\partial z_{ij_n}^l}{\partial [\mathbf{w}_1, \ldots, \mathbf{w}_{m_l}]}$, where $\mathbf{H}_l$ is a block diagonal square matrix whose diagonal blocks are all the same. Then, we can slightly revise the computation of the Hessian matrix, and extend the algorithm for fully-connected layers to convolutional layers.

Note that the accumulated error of ultimate network output can be linearly bounded by layer-wise error as long as the model is feed-forward. Thus, L-OBS is a general pruning method and friendly with most of feed-forward neural networks whose layer-wise Hessian can be computed expediently with slight modifications. However, if models have sizable layers like ResNet-101, L-OBS may not be economical because of computational cost of Hessian, which will be studied in our future work.

## 4 Experiments

In this section, we verify the effectiveness of our proposed Layer-wise OBS (L-OBS) using various architectures of deep neural networks in terms of compression ratio (CR), error rate before retraining, and the number of iterations required for retraining to resume satisfactory performance. CR is defined as the ratio of the number of preserved parameters to that of original parameters, lower is better. We conduct comparison results of L-OBS with the following pruning approaches: 1) Randomly pruning, 2) OBD [12], 3) LWC [9], 4) DNS [11], and 5) Net-Trim [6]. The deep architectures used for experiments include: LeNet-300-100 [2] and LeNet-5 [2] on the MNIST dataset, CIFAR-Net[2] [24] on the CIFAR-10 dataset, AlexNet [25] and VGG-16 [3] on the ImageNet ILSVRC-2012 dataset. For experiments, we first well-train the networks, and apply various pruning approaches on networks to evaluate their performance. The retraining batch size, crop method and other hyper-parameters are under the same setting as used in LWC. Note that to make comparisons fair, we do not adopt any other pruning related methods like Dropout or sparse regularizers on MNIST. In practice, L-OBS can work well along with these techniques as shown on CIFAR-10 and ImageNet.

### 4.1 Overall Comparison Results

The overall comparison results are shown in Table 1. In the first set of experiments, we prune each layer of the well-trained LeNet-300-100 with compression ratios: 6.7%, 20% and 65%, achieving slightly better overall compression ratio (7%) than LWC (8%). Under comparable compression ratio, L-OBS has quite less drop of performance (before retraining) and lighter retraining compared with LWC whose performance is almost ruined by pruning. Classic pruning approach OBD is also compared though we observe that Hessian matrices of most modern deep models are strongly non-diagonal in practice. Besides relative heavy cost to obtain the second derivatives via the chain rule, OBD suffers from drastic drop of performance when it is directly applied to modern deep models.

To properly prune each layer of LeNet-5, we increase tolerable error threshold $\epsilon$ from relative small initial value to incrementally prune more parameters, monitor model performance, stop pruning and set $\epsilon$ until encounter the "pruning inflection point" mentioned in Section 3.4. In practice, we prune each layer of LeNet-5 with compression ratio: 54%, 43%, 6% and 25% and retrain pruned model with

Table 1: Overall comparison results. (For iterative L-OBS, err. after pruning regards the last pruning stage.)

| Method | Networks | Original error | CR | Err. after pruning | Re-Error | #Re-Iters. |
|---|---|---|---|---|---|---|
| Random | LeNet-300-100 | 1.76% | 8% | 85.72% | 2.25% | $3.50 \times 10^5$ |
| OBD | LeNet-300-100 | 1.76% | 8% | 86.72% | 1.96% | $8.10 \times 10^4$ |
| LWC | LeNet-300-100 | 1.76% | 8% | 81.32% | 1.95% | $1.40 \times 10^5$ |
| DNS | LeNet-300-100 | 1.76% | 1.8% | - | 1.99% | $3.40 \times 10^4$ |
| L-OBS | LeNet-300-100 | 1.76% | 7% | **3.10%** | 1.82% | **510** |
| L-OBS (iterative) | LeNet-300-100 | 1.76% | **1.5%** | 2.43% | 1.96% | **643** |
| OBD | LeNet-5 | 1.27% | 8% | 86.72% | 2.65% | $2.90 \times 10^5$ |
| LWC | LeNet-5 | 1.27% | 8% | 89.55% | 1.36% | $9.60 \times 10^4$ |
| DNS | LeNet-5 | 1.27% | **0.9%** | - | 1.36% | $4.70 \times 10^4$ |
| L-OBS | LeNet-5 | 1.27% | 7% | **3.21%** | 1.27% | **740** |
| L-OBS (iterative) | LeNet-5 | 1.27% | **0.9%** | 2.04% | 1.66% | **841** |
| LWC | CIFAR-Net | 18.57% | 9% | 87.65% | 19.36% | $1.62 \times 10^5$ |
| L-OBS | CIFAR-Net | 18.57% | 9% | **21.32%** | 18.76% | **1020** |
| DNS | AlexNet (Top-1 / Top-5 err.) | 43.30 / 20.08% | **5.7%** | - | 43.91 / 20.72% | $7.30 \times 10^5$ |
| LWC | AlexNet (Top-1 / Top-5 err.) | 43.30 / 20.08% | 11% | 76.14 / 57.68% | 44.06 / 20.64% | $5.04 \times 10^6$ |
| L-OBS | AlexNet (Top-1 / Top-5 err.) | 43.30 / 20.08% | 11% | **50.04 / 26.87%** | 43.11 / 20.01% | $\mathbf{1.81 \times 10^4}$ |
| DNS | VGG-16 (Top-1 / Top-5 err.) | 31.66 / 10.12% | 7.5% | - | 63.38 / 38.69% | $1.07 \times 10^6$ |
| LWC | VGG-16 (Top-1 / Top-5 err.) | 31.66 / 10.12% | 7.5% | 73.61 / 52.64% | 32.43 / 11.12% | $2.35 \times 10^7$ |
| L-OBS (iterative) | VGG-16 (Top-1 / Top-5 err.) | 31.66 / 10.12% | 7.5% | **37.32 / 14.82%** | 32.02 / 10.97% | $\mathbf{8.63 \times 10^4}$ |

much fewer iterations compared with other methods (around $1:1000$). As DNS retrains the pruned network after every pruning operation, we are not able to report its error rate of the pruned network before retraining. However, as can be seen, similar to LWC, the total number of iterations used by DNS for rebooting the network is very large compared with L-OBS. Results of retraining iterations of DNS are reported from [11] and the other experiments are implemented based on TensorFlow [26]. In addition, in the scenario of requiring high pruning ratio, L-OBS can be quite flexibly adopted to an iterative version, which performs pruning and light retraining alternatively to obtain higher pruning ratio with relative higher cost of pruning. With two iterations of pruning and retraining, L-OBS is able to achieve as the same pruning ratio as DNS with much lighter total retraining: 643 iterations on LeNet-300-100 and 841 iterations on LeNet-5.

Regarding comparison experiments on CIFAR-Net, we first well-train it to achieve a testing error of 18.57% with Dropout and Batch-Normalization. We then prune the well-trained network with LWC and L-OBS, and get the similar results as those on other network architectures. We also observe that LWC and other retraining-required methods always require much smaller learning rate in retraining. This is because representation capability of the pruned networks which have much fewer parameters is damaged during pruning based on a principle that number of parameters is an important factor for representation capability. However, L-OBS can still adopt original learning rate to retrain the pruned networks. Under this consideration, L-OBS not only ensures a warm-start for retraining, but also finds important connections (parameters) and preserve capability of representation for the pruned network instead of ruining model with pruning.

Regarding AlexNet, L-OBS achieves an overall compression ratio of 11% without loss of accuracy with 2.9 hours on 48 Intel Xeon(R) CPU E5-1650 to compute Hessians and 3.1 hours on NVIDIA Tian X GPU to retrain pruned model (i.e. 18.1K iterations). The computation cost of the Hessian inverse in L-OBS is negligible compared with that on heavy retraining in other methods. This claim can also be supported by the analysis of time complexity. As mentioned in Section 3.4, the time complexity of calculating $\mathbf{H}_l^{-1}$ is $O\left(nm_{l-1}^2\right)$. Assume that neural networks are retrained via SGD, then the approximate time complexity of retraining is $O\left(IdM\right)$, where $d$ is the size of the mini-batch, $M$ and $I$ are the total numbers of parameters and iterations, respectively. By considering that $M \approx \sum_{l=1}^{l=L}\left(m_{l-1}^2\right)$, and retraining in other methods always requires millions of iterations ($Id \gg n$) as shown in experiments, complexity of calculating the Hessian (inverse) in L-OBS is quite economic. More interestingly, there is a trade-off between compression ratio and pruning (including retraining) cost. Compared with other methods, L-OBS is able to provide fast-compression: prune AlexNet to 16% of its original size without substantively impacting accuracy (pruned top-5 error 20.98%) even without any retraining. We further apply L-OBS to VGG-16 that has 138M parameters. To achieve more promising compression ratio, we perform pruning and retraining alteratively twice. As can be seen from the table, L-OBS achieves an overall compression ratio of 7.5% without loss

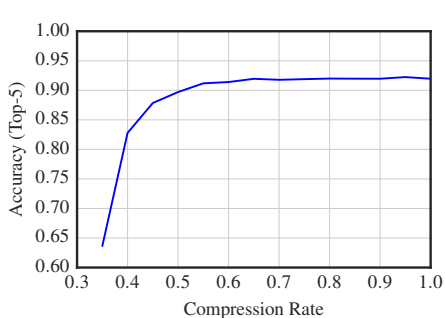
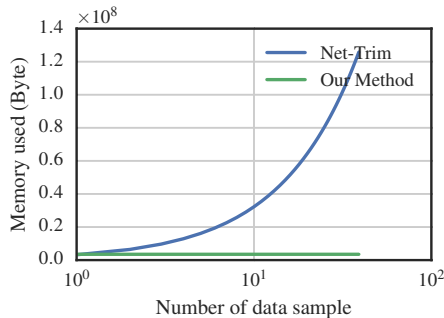

(a) Top-5 test accuracy of L-OBS on ResNet-50 under different compression ratios.

(b) Memory Comparion between L-OBS and Net-Trim on MNIST.

Table 2: Comparison of Net-Trim and Layer-wise OBS on the second layer of LeNet-300-100.

| Method | $\xi_r^2$ | Pruned Error | CR | Method | $\xi_r^2$ | Pruned Error | CR |
|---|---|---|---|---|---|---|---|
| Net-Trim | 0.13 | 13.24% | 19% | Net-Trim | 0.62 | 28.45% | 7.4% |
| L-OBS | 0.70 | 11.34% | 3.4% | L-OBS | 0.37 | 4.56% | 7.4% |
| L-OBS | 0.71 | 10.83% | 3.8% | Net-Trim | 0.71 | 47.69% | 4.2% |

of accuracy taking 10.2 hours in total on 48 Intel Xeon(R) CPU E5-1650 to compute the Hessian inverses and 86.3K iterations to retrain the pruned model.

We also apply L-OBS on ResNet-50 [27]. From our best knowledge, this is the first work to perform pruning on ResNet. We perform pruning on all the layers: All layers share a same compression ratio, and we change this compression ratio in each experiments. The results are shown in Figure 2(a). As we can see, L-OBS is able to maintain ResNet's accuracy (above 85%) when the compression ratio is larger than or equal to 45%.

## 4.2 Comparison between L-OBS and Net-Trim

As our proposed L-OBS is inspired by Net-Trim, which adopts $\ell_1$-norm to induce sparsity, we conduct comparison experiments between these two methods. In Net-Trim, networks are pruned by formulating layer-wise pruning as a optimization: $\min_{\mathbf{W}_l} \|\mathbf{W}_l\|_1$ s.t. $\|\sigma(\mathbf{W}_l^\top \mathbf{Y}^{l-1}) - \mathbf{Y}^l\|_F \leq \xi^l$, where $\xi^l$ corresponds to $\xi_r^l\|\mathbf{Y}^l\|_F$ in L-OBS. Due to memory limitation of Net-Trim, we only prune the middle layer of LeNet-300-100 with L-OBS and Net-Trim under the same setting. As shown in Table 2, under the same pruned error rate, CR of L-OBS outnumbers that of the Net-Trim by about six times. In addition, Net-Trim encounters explosion of memory and time on large-scale datasets and large-size parameters. Specifically, space complexity of the positive semidefinite matrix $Q$ in quadratic constraints used in Net-Trim for optimization is $O\left(2nm_l^2 m_{l-1}\right)$. For example, $Q$ requires about 65.7Gb for 1,000 samples on MNIST as illustrated in Figure 2(b). Moreover, Net-Trim is designed for multi-layer perceptrons and not clear how to deploy it on convolutional layers.

## 5  Conclusion

We have proposed a novel L-OBS pruning framework to prune parameters based on second order derivatives information of the layer-wise error function and provided a theoretical guarantee on the overall error in terms of the reconstructed errors for each layer. Our proposed L-OBS can prune considerable number of parameters with tiny drop of performance and reduce or even omit retraining. More importantly, it identifies and preserves the real important part of networks when pruning compared with previous methods, which may help to dive into nature of neural networks.

## Acknowledgements

This work is supported by NTU Singapore Nanyang Assistant Professorship (NAP) grant M4081532.020, Singapore MOE AcRF Tier-2 grant MOE2016-T2-2-060, and Singapore MOE AcRF Tier-1 grant 2016-T1-001-159.

## Footnotes

[1]For simplicity in presentation, we suppose the neural network is a feed-forward (fully-connected) network. In Section 3.4, we will show how to extend our method to filter layers in Convolutional Neural Networks.

[2] A revised AlexNet for CIFAR-10 containing three convolutional layers and two fully connected layers.

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
