[Supplementary Material · appendix.pdf]

# APPENDIX

## Proof of Theorem 3.2

We prove Theorem 3.2 via induction. First, for $l=1$, (8) holds as a special case of (2). Then suppose that Theorem 3.2 holds up to layer $l$:

$$\tilde{\varepsilon}^l \leq \sum_{h=1}^{l-1} \left( \prod_{k=h+1}^{l} \|\hat{\boldsymbol{\Theta}}_k\|_F \sqrt{\delta E^h} \right) + \sqrt{\delta E^l} \tag{10}$$

In order to show that (10) holds for layer $l+1$ as well, we refer to $\hat{\mathbf{Y}}^{l+1} = \sigma(\hat{\mathbf{W}}_{l+1}^\top \mathbf{Y}^l)$ as 'layer-wise pruned output', where the input $\mathbf{Y}^l$ is fixed as the same as the originally well-trained network not an accumulated input $\tilde{\mathbf{Y}}^l$, and have the following theorem.

**Theorem 5.1.** *Consider layer $l+1$ in a pruned deep network, the difference between its accumulated pruned output, $\tilde{\mathbf{Y}}^{l+1}$, and layer-wise pruned output, $\hat{\mathbf{Y}}^{l+1}$, is bounded by:*

$$\|\tilde{\mathbf{Y}}^{l+1} - \hat{\mathbf{Y}}^{l+1}\|_F \leq \sqrt{n}\|\hat{\boldsymbol{\Theta}}^{l+1}\|_F \tilde{\varepsilon}^l. \tag{11}$$

*Proof sketch:* Consider one arbitrary element of the layer-wise pruned output $\hat{\mathbf{Y}}^{l+1}$:

$$\begin{aligned}
\hat{y}_{ij}^{l+1} &= \sigma(\hat{\mathbf{w}}_i^\top \tilde{\mathbf{y}}_j^l + \hat{\mathbf{w}}_i^\top (\mathbf{y}_j^l - \tilde{\mathbf{y}}_j^l)) \\
&\leq \tilde{\mathbf{y}}_{ij}^{l+1} + \sigma(\hat{\mathbf{w}}_i^\top (\mathbf{y}_j^l - \tilde{\mathbf{y}}_j^l)) \\
&\leq \tilde{\mathbf{y}}_{ij}^{l+1} + |\hat{\mathbf{w}}_i^\top (\mathbf{y}_j^l - \tilde{\mathbf{y}}_j^l)|,
\end{aligned}$$

where $\hat{\mathbf{w}}_i$ is the $i$-th column of $\hat{\mathbf{W}}_{l+1}$. The first inequality is obtained because we suppose the activation function $\sigma(\cdot)$ is ReLU. Similarly, it holds for accumulated pruned output:

$$\tilde{y}_{ij}^{l+1} \leq \hat{y}_{ij}^{l+1} + |\hat{\mathbf{w}}_i^\top (\mathbf{y}_j^l - \tilde{\mathbf{y}}_j^l)|.$$

By combining the above two inequalities, we have

$$|\tilde{y}_{ij}^{l+1} - \hat{y}_{ij}^{l+1}| \leq |\hat{\mathbf{w}}_i^\top (\mathbf{y}_j^l - \tilde{\mathbf{y}}_j^l)|,$$

and thus have the following inequality in a form of matrix,

$$\|\tilde{\mathbf{Y}}^{l+1} - \hat{\mathbf{Y}}^{l+1}\|_F \leq \|\hat{\mathbf{W}}_{l+1}(\mathbf{Y}^l - \tilde{\mathbf{Y}}^l)\|_F \leq \|\hat{\boldsymbol{\Theta}}^{l+1}\|_F \|\mathbf{Y}^l - \tilde{\mathbf{Y}}^l\|_F$$

As $\tilde{\varepsilon}^l$ is defined as $\tilde{\varepsilon}^l = \frac{1}{\sqrt{n}}\|\mathbf{Y}^l - \tilde{\mathbf{Y}}^l\|_F$, we have

$$\|\tilde{\mathbf{Y}}^{l+1} - \hat{\mathbf{Y}}^{l+1}\|_F \leq \sqrt{n}\|\hat{\boldsymbol{\Theta}}^{l+1}\|_F \tilde{\varepsilon}^l.$$

This completes the proof of Theorem 11.

By using (2) ,(11) and the triangle inequality, we are now able to extend (10) to layer $l+1$:

$$\begin{aligned}
\tilde{\varepsilon}^{l+1} = \frac{1}{\sqrt{n}}\|\tilde{\mathbf{Y}}^{l+1} - \mathbf{Y}^{(l+1)}\|_F &\leq \frac{1}{\sqrt{n}}\|\tilde{\mathbf{Y}}^{l+1} - \hat{\mathbf{Y}}^{(l+1)}\|_F + \frac{1}{\sqrt{n}}\|\hat{\mathbf{Y}}^{l+1} - \mathbf{Y}^{(l+1)}\|_F \\
&\leq \sum_{h=1}^{l} \left( \prod_{k=h+1}^{l+1} \|\hat{\boldsymbol{\Theta}}^{k+1}\|_F \cdot \sqrt{\delta E^h} \right) + \sqrt{\delta E^{l+1}}.
\end{aligned}$$

Finally, we prove that (10) holds up for all layers, and Theorem 3.2 is a special case when $l=L$.

## Extensive Experiments and Details

### Redundancy of Networks

LeNet-300-100 is a classical feed-forward network, which has three fully connected layers, with 267K learnable parameters. LeNet-5 is a convolutional neural network that has two convolutional

Figure 2: Test accuracy on MNIST using LeNet-300-100 when continually pruning the first layer until pruning ratio is 100%. Comparison on ability to preserve prediction between LWC, ApoZ and our proposed L-OBS.

Figure 3: Distribution of sensitivity of parameters in LeNet-300-100's first layer. More than 90% of parameters' sensitivity scores are smaller than 0.001.

layers and two fully connected layers, with 431K learnable parameters. CIFAR-Net is a revised AlexNet for CIFAR-10 containing three convolutional layers and two fully connected layers.

We first validate the redundancy of networks and the ability of our proposed Layer-wise OBS to find parameters with the smallest sensitivity scores with LeNet-300-100 on MINIST. In all cases, we first get a well-trained network without dropout or regularization terms. Then, we use four kinds of pruning criteria: Random, LWC [9], ApoZW, and Layer-wise OBS to prune parameters, and evaluate performance of the whole network after performing every 100 pruning operations. Here, LWC is a magnitude-based criterion proposed in [9], which prunes parameters based on smallest absolute values. ApoZW is a revised version of ApoZ [16], which measures the importance of each parameter $\mathbf{W}_{lij}$ in layer $l$ via $\tau_{ij}^l = |\frac{1}{n}\sum_{p=1}^{n}(\mathbf{y}_{ip}^{l-1} \times \mathbf{W}_{lij})|$. In this way, both magnitude of the parameter and its inputs are taken into consideration.

Originally well-trained model LeNet-300-100 achieves 1.8% error rate on MNIST without dropout. Four pruning criteria are respectively conducted on the well-trained model's first layer which has 235K parameters by fixing the other two layers' parameters, and test accuracy of the whole network is recorded every 100 pruning operations without any retraining. Overall comparison results are summarized in Figure 2.

We also visualize the distribution of parameters' sensitivity scores $L_q$'s estimated by Layer-wise OBS in Figure 3, and find that parameters of little impact on the layer output dominate. This further verifies our hypothesis that deep neural networks usually contain a lot of redundant parameters. As shown in the figure, the distribution of parameters' sensitivity scores in Layer-wise OBS are heavy-tailed. This means that a lot of parameters can be pruned with minor impact on the prediction outcome.

Figure 4: Retraining pattern of LWC and L-OBS. L-OBS has a better start point and totally resume original performance after 740 iterations for LeNet-5.

Random pruning gets the poorest result as expected but can still preserve prediction accuracy when the pruning ratio is smaller than 30%. This also indicates the high redundancy of the network.

Compared with LWC and ApoZW, L-OBS is able to preserve original accuracy until pruning ratio reaches about 96% which we call as "pruning inflection point". As mentioned in Section 3.4, the reason on this "pruning inflection point" is that the distribution of parameters' sensitivity scores is heavy-tailed and sensitivity scores after "pruning inflection point" would be considerable all at once. The percentage of parameters with sensitivity smaller than 0.001 is about 92% which matches well with pruning ratio at inflection point.

L-OBS can not only preserve models' performance when pruning one single layer, but also ensures tiny drop of performance when pruning all layers in a model. This claim holds because of the theoretical guarantee on the overall prediction performance of the pruned deep neural network in terms of reconstructed errors for each layer in Section 3.3. As shown in Figure 4, L-OBS is able to resume original performance after 740 iterations for LeNet-5 with compression ratio of 7%.

**How To Set Tolerable Error Threshold**

One of the most important bounds we proved is that there is a theoretical guarantee on the overall prediction performance of the pruned deep neural network in terms of reconstructed errors for each pruning operation in each layer. This bound enables us to prune a whole model layer by layer without concerns because the accumulated error of ultimate network output is bounded by the weighted sum of layer-wise errors. As long as we control layer-wise errors, we can control the accumulated error.

Although L-OBS allows users to control the accumulated error of ultimate network output $\tilde{\varepsilon}^L = \frac{1}{\sqrt{n}}\|\tilde{\mathbf{Y}}^l - \mathbf{Y}^l\|_F$, this error is used to measure difference between network outputs before and after pruning, and is not strictly inversely proportional to the final accuracy. In practice, one can increase tolerable error threshold $\epsilon$ from a relative small initial value to incrementally prune more and more parameters to monitor model performance, and make a trade-off between compression ratio and performance drop. The corresponding relation (in the first layer of LeNet-300-100) between the tolerable error threshold and the pruning ratio is shown in Figure 5.

**Iterative Layer-wise OBS**

As mentioned in Section 4.1, to achieve better compression ratio, L-OBS can be quite flexibly adopted to its iterative version, which performs pruning and light retraining alternatively. Specifically, the two-stage iterative L-OBS applied to LeNet-300-100, LeNet-5 and VGG-16 in this work follows the following work flow: pre-train a well-trained model → prune model → retrain the model and reboot performance in a degree → prune again → lightly retrain model. In practice, if required compression ratio is beyond the "pruning inflection point", users have to deploy iterative L-OBS though ultimate compression ratio is not of too much importance. Experimental results are shown in Tabel 3, 4 and 5,

Figure 5: The corresponding relation between tolerable error threshold and pruning ratio.

where CR(n) means ratio of the number of preserved parameters to the number of original parameters after the $n$-th pruning.

Table 3: For LeNet-300-100, iterative L-OBS(two-stage) achieves compression ratio of 1.5%

| Layer | Weights | CR1 | CR2 |
|-------|---------|-----|-----|
| fc1 | 235K | 7% | 1% |
| fc2 | 30K | 20% | 4% |
| fc3 | 1K | 70% | 54% |
| Total | 266K | 8.7% | 1.5% |

Table 4: For LeNet-5, iterative L-OBS(two-stage) achieves compression ratio of 0.9%

| Layer | Weights | CR1 | CR2 |
|-------|---------|-----|-----|
| conv1 | 0.5K | 60% | 20% |
| conv2 | 25K | 60% | 1% |
| fc1 | 400K | 6% | 0.9% |
| fc2 | 5K | 30% | 8% |
| Total | 431K | 9.5% | 0.9% |

Table 5: For VGG-16, iterative L-OBS(two-stage) achieves compression ratio of 7.5%

| Layer | conv1_1 | conv1_2 | conv2_1 | conv2_2 | conv3_1 | conv3_2 | conv3_3 | conv4_1 |
|-------|---------|---------|---------|---------|---------|---------|---------|---------|
| Weights | 2K | 37K | 74K | 148K | 295K | 590K | 590K | 1M |
| CR1 | 70% | 50% | 70% | 70% | 60% | 60% | 60% | 50% |
| CR2 | 58% | 36% | 42% | 32% | 53% | 34% | 39% | 43% |

| Layer | conv4_2 | conv4_3 | conv5_1 | conv5_2 | conv5_3 | fc6 | fc7 | fc8 |
|-------|---------|---------|---------|---------|---------|-----|-----|-----|
| Weights | 2M | 2M | 2M | 2M | 2M | 103M | 17M | 4M |
| CR1 | 50% | 50% | 70% | 70% | 60% | 8% | 10% | 30% |
| CR2 | 24% | 30% | 35% | 43% | 32% | 2% | 5% | 17% |