[Reviews · NeurIPS 2017]

Reviewer 1



The authors consider neural network pruning problem. They approach the problem by pruning the network layer by layer. The paper has contributions in multiple directions both algorithmic and theoretical. Theoretically they obtain the pruning error between the pruned network and the original (trained baseline) network. Since the approach is layer by layer, the output error is also in terms of the accumulation of the individual layers. While theoretical results are intuitive and clean, I am not sure how novel they are. It appears to me that [12, 13] already did similar analysis for shallow networks and the observation of this work is propagating the layerwise errors for deep networks. Authors should address this (in particular Lemma 3.1 and Theorem 3.3) as it is important to determine the novelty of the work. Algorithmically, they propose a fast algorithm for inverse Hessian calculation and essentially implement a careful deep version of the optimal brain surgeon. The proposed algorithm outperforms competing methods (closest one is DNS) and requires less training which are both plus. I recommend the authors additionally compare their methods to iterative-hard thresholding methods (on top of l1 regularization (nettrim) such as Jin et al. "Training Skinny Deep Neural Networks with Iterative Hard Thresholding Methods". Secondly, is there any argument for DNS and the proposed methods are performing the best? Can authors provide an argument?

Reviewer 2



Summary: This paper adapts Optimal Brain Surgeon (OBS) method to a local version, and modified the objective function to be the target activation per each layer. Similar to OBS, it uses an approximation to compute Hessian inverse by running through the dataset once. Compare to prior methods, it finishes compression with much less retraining iterations. A theoretical bound on the total error based on local reconstruction error is provided. Pros: - The paper explores a local version of OBS and shows effectiveness of proposed method in terms of less time cost for retraining the pruned network. - Leveraging local second order information is a much clever way than solving/retraining a network through brute force. - The proposed method achieves good compression ratio without losing performance, with much less retraining iterations. Comments: - I recommend the authors to include the basics of Optimal Brain Surgeon (OBS) as a background section. This will make the flow of the paper much better. Many of the techniques used later (e.g. Lagrange multiplier of the MSE error function, iterative algorithm for computing the Hessian inverse) should be part of the background. - The formulation of layer-wise objective is difficult to understand. Z^l is also a function of Theta^l, and since Z^l = Y^(l-1) W^l, then E^l is actually not even a function of Theta^l, since both terms cancel out in Z^l and hat(Z)^l. This is another reason why the - paper should have included OBS in the background, since here it uses Z^l as a local target activation value, and no longer treated as a function of Theta^l. I had read section 3.2 over many times to realize this point. - Line 129: “For a well-trained network, gradient w.r.t. Parameters is small enough to be ignored as well”. This is not true. For the original OBS, it is true since the error is the total loss function of the network. For a local MSE function, the gradient of E^l wrt. Theta^l is literally zero (not just small enough). - I recommend the authors move the proof Theorem 3.2 to the Appendix. This bound is good to know (that the total error is bounded by each local errors) but not very useful and hurts the flow of the paper. - I recommend the authors edit section 3.4 into algorithmic format. It is hard to read an algorithm in a paragraph. Same for section “prune convolutional layer”. - Line 191: Again, I don’t understand why “well-trained” is part of the clause here. For any function, if you perturb the input a little bit, the output will be quite close. In the original OBS, it is the final output vs. the target, so “well-trained” is necessary. But here, the target is itself. - The paper gives some asymptotic bound on time complexities, but I am still curious of the actual wall clock timing compared to related work (DNS, Net-Trim), especially the time cost to compute the Hessian inverse. - The paper claims that Net-Trim has memory limitation. But maybe it is worth trying a mini-batch version of Net-Trim? - The paper puts ResNet-101 as future work. It would have been better if the authors can give preliminary results to give insights on how expensive it is to compute Hessian inverse on a deeper network like this. Based on the time complexity, I don’t see why running ResNet-101 with L-OBS is an issue. - The model achieves good results with much less number of retraining iterations. However, in the AlexNet experiment, it is still losing to DNS in terms of compression ratio. Although, the paper claims to have faster compression time, one major use case for network compression/sparsification deployment on embedded systems, in which the compression ratio is more of a concern than time cost to retrain the network. For completeness, I also recommend the authors to implement DNS and report numbers on VGG-16. Overall, I feel the paper has some good contribution, but the paper clarity needs to be improved by a lot (Background section, local error function formulation). Some experiments need to be clarified (e.g. wall clock time comparison), and completed (DNS numbers on VGG). Based on these points, I recommend marginal accept, conditioned on a major revision. After rebuttal note: I am glad that DNS numbers on VGG and ResNet is provided. After seeing the rebuttal I decided to upvote my rating to 7. I hope the authors will reorganize the paper as promised in the rebuttal.

Reviewer 3



The paper extends the optimal brain surgeon approach to pruning parameters by defining a layerwise loss to determine which parameters to prune. The loss computes the change in the function of the layer and a 2nd order is used to approximate it. This objective is minimized with a Lagrangian constraint to find the the best parameters to prune. The algorithm is well-reasoned but the claim in the abstract that "there is a guarantee that one only needs to perform a light retraining process" is overstated. The theoretical results only bound the error at each layer and say nothing of the end-to-end classification error. Though experimentally, it is observed that the algorithm requires less retraining - but without the proper bounds that does not constitute a guarantee. The method is evaluated on an array of datasets inclusing MNIST, CIFAR and Imagenet. It achieves significant compression rates with comparatively smaller drops in performance. Typos: L147 "By far" -> "So far"